# Aging-associated sinus arrest and sick sinus syndrome in adult zebrafish

**Jianhua Yan[1,2,3], Hongsong Li[1,2], Haisong Bu[1,2], Kunli Jiao[3], Alex X. Zhang[1,2], Tai Le[4], Hung Cao[4,5], Yigang Li[3], Yonghe Ding[1,2]\*, Xiaolei Xu[1,2]\***

1 Department of Biochemistry and Molecular Biology, Mayo Clinic, Rochester, Minnesota, United States of America, 2 Department of Cardiovascular Medicine, Mayo Clinic, Rochester, Minnesota, United States of America, 3 Division of Cardiology, Xinhua Hospital Affiliated To Shanghai Jiaotong University School Of Medicine, Shanghai, China, 4 Department of Electrical Engineering and Computer Science, UC Irvine, Irvine, California, 5 Department of Biomedical Engineering, UC Irvine, Irvine, California

\* xu.xiaolei@mayo.edu (XX); ding.yonghe@mayo.edu (YD)

**Data Availability Statement:** All relevant data are within the paper.

**Funding:** This work was supported by grants from the National Institutes of Health (NIH) (https://www.nih.gov/) R01-HL107304, R01-HL081753, R01-GM63904 and the Mayo Foundation

## Abstract

Because of its powerful genetics, the adult zebrafish has been increasingly used for studying cardiovascular diseases. Considering its heart rate of ~100 beats per minute at ambient temperature, which is very close to human, we assessed the use of this vertebrate animal for modeling heart rhythm disorders such as sinus arrest (SA) and sick sinus syndrome (SSS). We firstly optimized a protocol to measure electrocardiogram in adult zebrafish. We determined the location of the probes, implemented an open-chest microsurgery procedure, measured the effects of temperature, and determined appropriate anesthesia dose and time. We then proposed an PP interval of more than 1.5 seconds as an arbitrary criterion to define an SA episode in an adult fish at ambient temperature, based on comparison between the current definition of an SA episode in humans and our studies of candidate SA episodes in aged wild-type fish and *Tg(SCN5A-D1275N)* fish (a fish model for inherited SSS). With this criterion, a subpopulation of about 5% wild-type fish can be considered to have SA episodes, and this percentage significantly increases to about 25% in 3-year-old fish. In response to atropine, this subpopulation has both common SSS phenotypic traits that are shared with the *Tg(SCN5A-D1275N)* model, such as bradycardia; and unique SSS phenotypic traits, such as increased QRS/P ratio and chronotropic incompetence. In summary, this study defined baseline SA and SSS in adult zebrafish and underscored use of the zebrafish as an alternative model to study aging-associated SSS.

## 1. Introduction

A mammalian heart strictly controls the rhythm of the beat to meet the demand of the body for circulation. Heart rhythm is primarily initiated by the automatic beating of pacemaker cells located in the sinoatrial node (SAN), a specialized area in the upper right chamber of a mammalian heart. The initial innate electrical potential transmits from the SAN to the atrioventricular node (AVN) and finally passes to the His-Purkinje system. This well-controlled rhythmic contraction is modulated both positively by sympathetic nerves and negatively by parasympathetic

FP00093430 (https://www.mayoclinic.org/) to XX, the Ted and Loretta Rogers Cardiovascular Career Development Award Honoring Hugh C. Smith from Mayo Clinic (FP00093430) to YD, and the National Science Foundation (NSF) (https://www.nsf.gov/funding/) #1652818 and NIH R41-OD024874 to HC. The funders had no role in study design, data collection and analysis, decision to publish, or preparation of the manuscript.

**Competing interests:** The authors have declared that no competing interests exist.

nerves[1]. Sick sinus syndrome (SSS) is one type of heart rhythm disorder that affects the sino-atrial node; the disorder is also called sinus node dysfunction (SND) [2]. The causes of SSS can be divided into both intrinsic and extrinsic factors that disrupt the SAN function. Intrinsic causes include age-related degenerative fibrosis, ion channel dysfunction, and remodeling of the SAN. Extrinsic factors include the use of certain pharmacologic agents, metabolic disturbances, and autonomic dysfunctions that exacerbate SSS. A critical index for diagnosing SSS is sinus arrest (SA), which can be determined with surface electrocardiography (ECG). In humans, a PP or RR interval more than 1.6 seconds is considered a candidate episode for SA, more than 2.0 seconds is widely accepted as a definitive SA episode [3], and more than 3.0 seconds is considered a criterion for implanting a pacemaker [4]. Other phenotypic traits for SSS include sinus bradycardia, sino-atrial exit block, tachycardia-bradycardia syndrome, decreased P amplitude, increased QRS/P ratio, dizziness, and syncope [5–10]. SSS is the major reason for installation of a pacemaker and the common cause of sudden cardiac death [11]. Aging is a well-known risk factor for SSS, and the prevalence of SSS in elderly people can reach 1 in 600 [2, 12], likely due to age-related degenerative fibrosis of the sinus node [2, 13]. Although genetic contributions to SSS have been recognized, only a few genetic factors have been identified. Sodium voltage-gated channel alpha subunit 5 (*SCN5A*) and hyperpolarization-activated cyclic nucleotide-gated channels 4 (*HCN4*) are two well-established disease causative genes for SSS [14, 15].

The zebrafish (*Danio rerio*) is emerging as a powerful vertebrate model for studying cardiovascular diseases, largely because of its amenability to large scale genetic studies, enabling rapid discovery of new genetic factors [16]. The electric wiring system in a zebrafish heart has remarkable similarities to that in a mammalian heart. Its sinoatrial node is a ring-like structure between the sinus venosus and the atrium [17, 18], and the node seems to use pacemaking mechanisms similar to those in mammals [19]. The heart is innervated and controlled by both sympathetic and parasympathetic nerve fibers [18]. Single-channel surface ECG technology has shown that the zebrafish has heart rate (HR) and ECG patterns similar to those in humans [20]. Electrophysiologic indices, including shape and duration of action potential, resemble those in humans more than those in the mouse [16, 21–23]. A spectrum of rhythmic diseases has been successfully modeled in zebrafish, including long and short QT-interval syndrome and atrial fibrillation [24]. The *SCN5A* gene encodes the α subunit of the cardiac sodium (Na$^+$) channel, Na$_V$1.5. A missense mutation, D1275N, in the *SCN5A* gene is associated with cardiac conduction disease, sinus node dysfunction, and arrhythmia [14]. Transgenic overexpression of *SCN5A-D1275N* in zebrafish led to bradycardia, conduction-system abnormalities and premature death, which could be considered a zebrafish model for inherited SSS [25].

To further establish the zebrafish as a model for heart rhythm disorders, we describe the optimization of an ECG protocol using a commercial available ECG system for adult fish system. A series of experiments led us to propose a PP interval of more than 1.5 seconds as a criterion for an SA episode in adult zebrafish. With this criterion, we found that a small percentage of wild-type (WT) fish have SA episodes even when they are young, and the percentage increases significantly during the aging process. This subpopulation of fish manifests SSS phenotypic traits, such as bradycardia, increased QRS/P ratio, aberrant response to atropine, some of which are distinct from the *Tg(SCN5A-D1275N)* model. In summary, our study established zebrafish as an alternative vertebrate model for studying aging-associated SA and SSS.

## 2. Materials and methods

### 2.1 Zebrafish husbandry

Adult WT zebrafish from three different strains (WIK, NIHGR, TU) were used for the study. All zebrafish were maintained in a controlled environment at a temperature of 28˚C with a

light-dark cycle of about 14 hours-10 hours. All animal study procedures were performed according to the *Guide for the Care and Use of Laboratory Animals* published by the US National Institutes of Health (NIH Publication No. 85–23, revised 1996) and the Mayo Clinic Institutional Animal Care and Use Committee (approved protocol number A17610).

## 2.2 Anesthesia

The anesthetic agent used in this study was tricaine, which is the only anesthetic approved for certain aquatic species by the US Food and Drug Administration and the most widely used sedative and anesthetic for zebrafish. The solution of pH 7.0-adjusted tricaine (MS-222, Sigma) at the typical anesthesia concentrations (200 mg/L, 0.02%) for zebrafish was dissolved in E3 medium (containing 5 mM NaCl, 0.17 mM KCl, 0.33 mM $CaCl_2$, and 0.33 mM $MgSO_4$) and was used to immobilize the fish.

## 2.3 Microsurgery to open the pericardium sac

Microsurgery has been reported to boost and to reduce background noise of the ECG signal [26, 27]. The operations were conducted typically 1 week before ECG. The zebrafish were anesthetized with 0.02% tricaine for 3 minutes before subjected to microsurgery on a dented sponge. With use of a dissecting microscope, tweezers were used to puncture the skin on the surface of the heart and to make a small incision through the skin. The silvery epithelial layer of the hypodermis was torn gently with the tip of the tweezers to expose the heart visually. After the surgery was completed, usually within 1–2 minutes, fish were returned to tank water for recovery. After 1 week, the surface skin wound was healed, but the pericardial sac was not; as a result, the cardiac electrical current could flow through with minimal attenuation.

## 2.4 Adult zebrafish ECG recording

Unless specified, ECG recording was performed at ambient temperature which was maintained at 24˚C. Before subjected to ECG recording, tanks holding adult fish were acclimated for about an hour after transferred from the circulating fish facility to the laboratory bench. When a different temperature other than ambient temperature is needed, a temperature-controlled chamber set-up was used. The temperature-controlled chamber was made by covering the ECG recording system with a foam box. The ECG machine was held on top of a heating plate controlled by a heating machine. Temperature within the chamber could range from 22˚C to 32˚C, as measured by a thermometer inside the chamber. ECG signals were then obtained with the ECG recording and analysis System, according to instructions (ZS-200, iWorx Systems, Inc). ECG signals were amplified and filtered at 0.5 Hz high pass and 200 Hz low pass. After being anesthetized with 0.02% tricaine for 6 minutes, the fish were transferred onto a dented sponge to stably hold them upside down. A few drops of E3 medium were added to the fish to maintain moisture on the fish surface. ECG signals were recorded for 2 minutes before the fish were transferred to E3 water for recovery.

## 2.5 Analysis of zebrafish ECG recording

A series of ECG variables, including HR, P-wave amplitude, R-wave amplitude, and PP and RR intervals were calculated with an in-house Matlab code [28]. The root mean square of the successive differences (RMSSD) [29] was used to evaluate the short-period (60 seconds) beat-

to-beat variance of HR according to the following formula:

$$\text{RMSSD (milliseconds)} = \sqrt{\frac{\sum_{i-1}^{n-1}(PPi + 1 - PPi)\ ^2}{n-1}}$$

### 2.6 Atropine administration

Unlike the conventional route of administration in mammals, atropine was administered by gill titration because of the robust blood circulation in the zebrafish gill. The fish were anesthetized and held upside down on a dented sponge while ECG was performed. After a reliable ECG signal was obtained, 4 µg/g atropine (A0132, Sigma) was added to the gill region of the fish. The dose of the atropine was calculated as the weight of drug that is normalized by the body weight of the fish, and the duration of the atropine administration was determined experimentally.

### 2.7 Cardiac function assay

Cardiac function was quantified with a Vevo 3100 high-frequency imaging system (FUJIFILM VisualSonics, Inc) equipped with a 50-MHz linear array transducer (MX700). Zebrafish were anesthetized with 0.02% tricaine for 6 minutes and held upside down in a dented sponge. The MX700 transducer was positioned above the zebrafish to align sagittally to the heart. Images were acquired and processed with the Vevo LAB workstation. Data were acquired and processed as previously described [27]. End-diastolic volume (EDV) and end-systolic volume (ESV) were calculated using the biplane area-length formula: V = 2/3AAL*LAL, in which AAL is the ventricle area in the transverse plane (short axis) and LAL is the ventricle length in the longitudinal plane (long axis), and ejection fraction (EF) = (EDV −ESV)/EDV. For each index, measurements were obtained during 3 to 5 independent cardiac cycles per fish to determine average values. With B-mode imaging, a Doppler gate (window) was positioned downstream from the atrioventricular valve in the ventricular inflow region to interrogate inflow velocities. Pulsed-wave Doppler signals were also recorded for 3 to 5 independent cardiac cycles per fish. Signals of passive E wave (peak velocity of early ventricular filling) and active A wave (peak velocity of late ventricular filling), isovolumic contraction time (IVCT), ejection time (ET), and isovolumic relaxation time (IVRT) were measured to determine the myocardial performance index (MPI), as follows: MPI = (IVCT + IVRT) / ET.

### 2.8 Statistics

Either an unpaired 2-tailed Student *t* test or a nonparametric test was used to compare 2 groups, depending on whether parametric assumptions of normality and equality of variance were met. A one-way analysis of variance (or Kruskal-Wallis) test followed by a post hoc Tukey test was used to compare 3 or more groups. The $\chi^2$ test was used for rate comparison. *P* values less than 0.05 were considered statistically significant. Results are presented as mean±SD. All statistical analyses were conducted with GraphPad Prism 6 software (GraphPad Software).

## 3. Results

### 3.1 Reliable ECG signals can be obtained after optimizing probe location, conducting microsurgery, and minimizing effects of temperature and anesthesia

While acquiring ECG signal in adult zebrafish, we noted that different ECG patterns can be obtained when the electrodes are placed at different locations. If the 2 electrodes spanned the

heart, an inverted QRS complex that is opposite to the P wave was obtained (Fig 1A, i). When the positive probe was aligned to the heart, typical P, QRS, and T waves were obtained (Fig 1A, ii). If the electrodes were moved further caudally, amplitudes of P, QRS, and T waves were

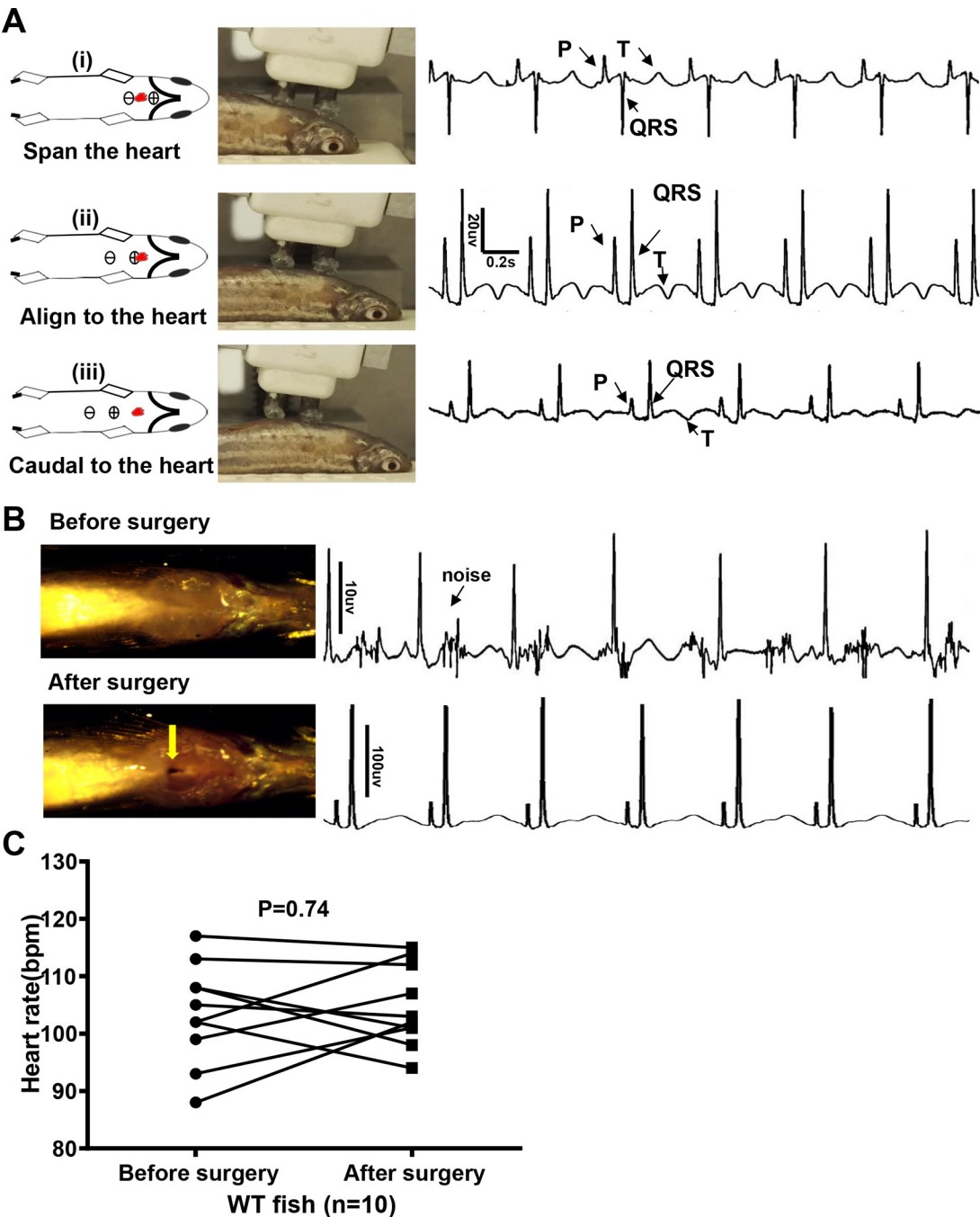

**Fig 1. Quality of electrocardiography (ECG) signals is improved by optimizing location of the probes and microsurgery.** A, Different ECG waveform with distinct P, R, S, and T waves when probes were at a different location relative to the heart (red dots). We recommend aligning the positive probe to the heart (ii), whereby the ECG waveform is similar to that in humans. B, Microsurgery was conducted to open the pericardium sac and thus eliminate noise and enhance amplitude of the waveforms by about 10-fold. C, Change in heart rate with surgery. bpm indicates beats per minute; WT, wild-type.

attenuated (Fig 1A, iii). Thus, we used the position of the positive probe aligned to the heart for later experiments.

An adult zebrafish is able to survive microsurgery that partially opens the pericardial sac to remove the silver-white membrane [30–33]. We found that the operation is also able to boost the ECG signal, as indicated by about 10-fold higher amplitudes of P, QRS, and T waves, accompanied by significantly reduced background noise (Fig 1B). Microsurgery has no significant impact on HR (Fig 1C).

The design of the ECG recording system we used allows an adult fish to be physically constrained without anesthesia for a short time, during which ECG signals with P wave and QRS complex can be documented despite a relatively noisy background caused by movement of the fish body and gills (Fig 2A). The average HR in WT fish at ambient temperature (ie, 24˚C) is about 100 beats per minutes (bpm). Because the zebrafish is an ectotherm that adjusts its body temperature according to its environment [26], we built a chamber for temperature control and plotted the change of HR as a function of chamber temperature. We found that the HR could change from 80 bpm at 22˚C to 180 bpm at 32˚C. Interestingly, we noted a higher beat to beat variability at lower temperature, e.g. between 22˚C and 26˚C, which warrants further quantification and follow up studies in the future (Fig 2A and 2B). A similar nearly linear HR-temperature relationship was noted when fish were anesthetized (Fig 2B). Because the HR could increase about 10 bpm per 1˚C, temperature control was needed to ensure consistent ECG data in adult fish.

Anesthetization of the fish with tricaine incubation effectively reduced the background noise caused by body and gill movement, which become stable at 6 minutes after anesthesia (Fig 2C). Moreover, tricaine initially increased HR significantly, which peaked at about 90 seconds, likely due to stress response and excessive excitement of sympathetic nerves. Then the HR gradually decreased to the baseline level (Fig 2D). Also, the incidence of HR variation was higher during the first minute of ECG documentation.

Accordingly, we recommend the following protocol for obtaining a reliable ECG signal with an iWorx system: 1) open the pericardial sac and remove the silver-white membrane 7 days before ECG recording; 2) put fish into 0.02% tricaine E3 water for 6 minutes at desired temperature; and 3) one minute after a fish is held on a dented sponge when more reliable ECG signals can be obtained, record ECG signals for additional 2 minutes (Fig 2E).

## 3.2 PP interval more than 1.5 seconds can be used to define an SA episode in zebrafish

Similar to waveforms in both mice and humans, a typical zebrafish ECG waveform consists of P, QRS, and T waves that are comparable to other previously published recordings (Fig 3A) [34–36]. The average HR in an adult zebrafish is about 100 bpm at ambient temperature, similar to the rate of 60 to 100 bpm in humans. In contrast, the average HR in mice is 450 to 600 bpm [34], which is significantly higher than that in humans and fish (Fig 3B). In contrast to a relatively stable HR in both humans and mice, HR variation after anesthesia was higher in adult zebrafish, as indicated by the significantly higher variation in either PP intervals or $PP_i$ / $PP_{i+1}$ (ratio of 2 adjacent PP intervals) (Fig 3B–3C). Therefore, the baseline HR variation in WT fish needs to be defined; this is the foundation for modeling related human rhythmic diseases such as SA and SSS. As a control, we included *Tg(SCN5A-D1275N)* in our study, the first zebrafish model for inherited SSS [37].

In humans, a PP or RR interval more than 1.6 seconds is considered a candidate episode for SA, and an interval more than 2.0 seconds is widely accepted as a definitive SA episode. Given that HR is 60 to 90 bpm in humans and 100 bpm in unanesthetized adult zebrafish at ambient

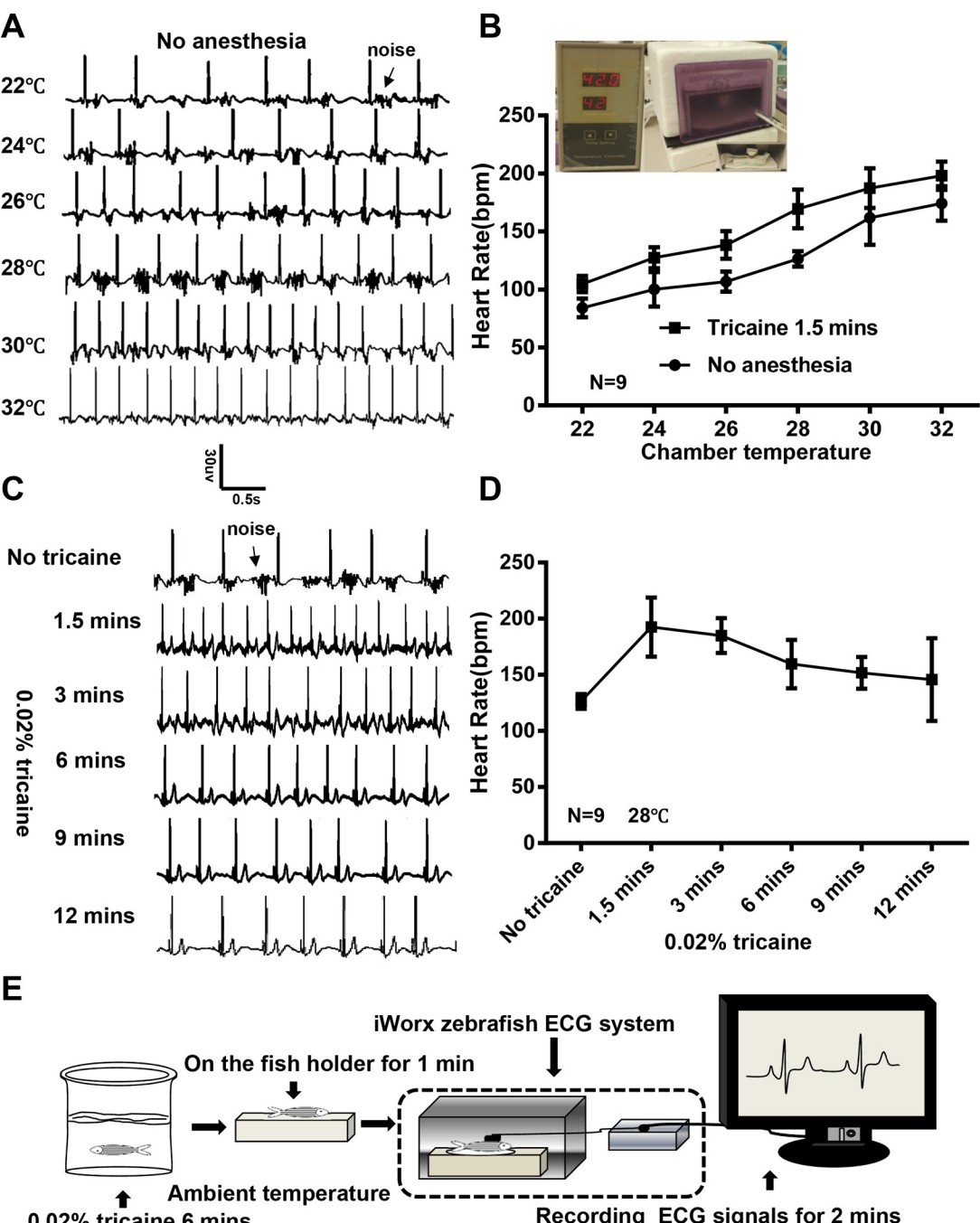

**Fig 2. Optimization of temperature and anesthesia for reliable ECG recording in adult zebrafish.** A, Representative electrocardiogram (ECG) for a single fish without anesthesia. Chamber temperature was changed from 22°C to 32°C. B, Heart rate significantly increased with increased chamber temperature, which is independent of anesthesia. Inset shows in-house temperature-controlled chamber. C, Representative ECG for a single fish before and after anesthethia. Temperature was controlled at 28°C. Noise caused by gill movement is reduced after 6 minutes of anesthesia. D, Heart rate to the time after anesthesia. E, Schematic of protocol for conducting ECG with the iWorx system. bpm indicates beats per minute.

temperature, or a 1.33-fold difference, we deduced that an abnormal PP or RR interval in fish is 1.2 to 1.6 seconds. We measured both PP and RR intervals in 59 WT fish at 1 year old and found these two parameters are similar. Thus we used the PP interval as the parameter to

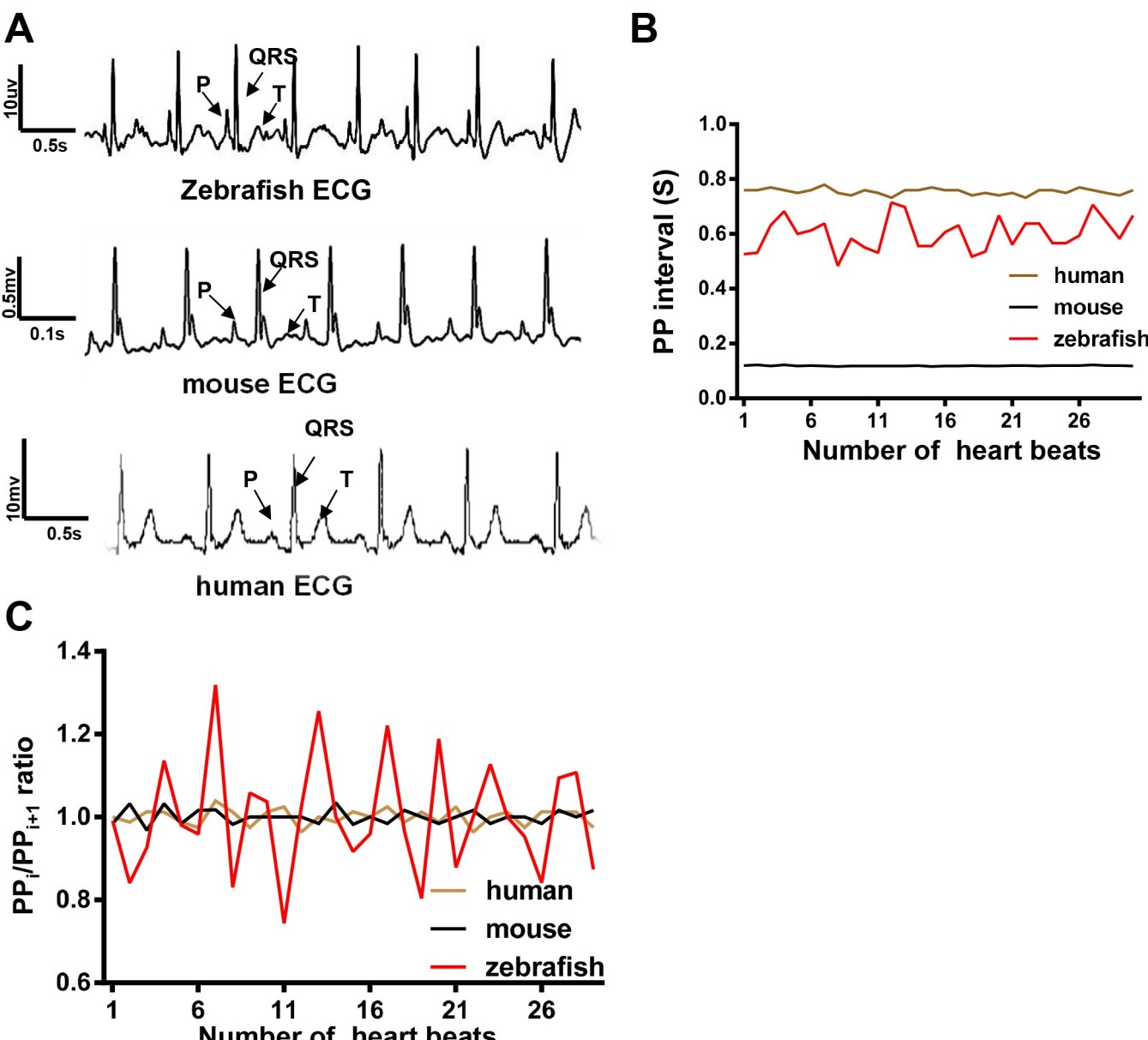

**Fig 3. Heart rate was more variable in zebrafish than in mice and humans under anesthesia.** A, Electrocardiographic (ECG) waveforms with distinguishable P, QRS, and T waves were similar in zebrafish, mice, and humans. B, Heart rate variation (HRV) was measured by quantifying average PP intervals every 30 beats in a representative fish, mouse, and human. HR in the zebrafish was similar to that in a human but was much higher than that in a mouse. HRV in zebrafish was larger than that in both a mouse and a human. C, HRV was measured by quantifying $PP_i/PP_{i+1}$ (ratio of 2 adjacent PP intervals) in a representative fish, mouse, and human. The ratio in a zebrafish was much larger than that in both a mouse and a human.

determine episode for SA. Out of these 59 WT fish, three had episodes of PP intervals more than 1.2 seconds, one of which was more than 1.6 seconds. In contrast, 10 of 16 *Tg (SCN5A-D1275N)* fish had episodes of PP intervals of more than 1.2 seconds, seven of which were more than 1.6 seconds (Fig 4A). We manually analyzed ECG signals for two WT fish and three *Tg(SCN5A-D1275N)* fish that were considered to have SA if 1.2 seconds was used as a cut-off but not if 1.6 seconds was used as a cut-off. Because of the high candidate episode number per minute in WT fish number 1 and 2 and *Tg(SCN5A-D1275N)* fish number 2, we

**A**

| Fish Type | Age | N | HR, bpm | Percentage of fish with long PP interval (No. of fish) | | | | |
|---|---|---|---|---|---|---|---|---|
| | | | | PP >1.2 s | PP >1.3 s | PP >1.4 s | PP >1.5 s | PP >1.6 s |
| WT | 1y | 59 | 103.3±15.7 | 5.1(3) | 5.1(3) | 5.1(3) | 5.1(3) | 3.4(1) |
| *Tg(SCN5A-D1275N)* | 1y | 16 | 73.8±14.9 | 62.5(10) | 62.5(10) | 62.5(10) | 50.0(8) | 43.8(7) |
| WT | 3y | 8 | 84.6±22.5 | 62.5(5) | 37.5(3) | 37.5(3) | 25(2) | 25(2) |

**B**

| fish | HR, bpm | PP >1.2 s epm | PP >1.3 s epm | PP >1.4 s epm | PP >1.5 s epm | PP >1.6 s epm | SA |
|---|---|---|---|---|---|---|---|
| WT-1 | 67 | 7 | 5 | 2 | 1 | 0 | y |
| WT-2 | 71 | 3 | 5 | 2 | 2 | 0 | y |
| *Tg(SCN5A-D1275N)-1* | 71 | 2 | 1 | 1 | 0 | 0 | n |
| *Tg(SCN5A-D1275N)-2* | 76 | 13 | 3 | 2 | 1 | 0 | y |
| *Tg(SCN5A-D1275N)-3* | 69 | 3 | 2 | 1 | 0 | 0 | n |

**C**

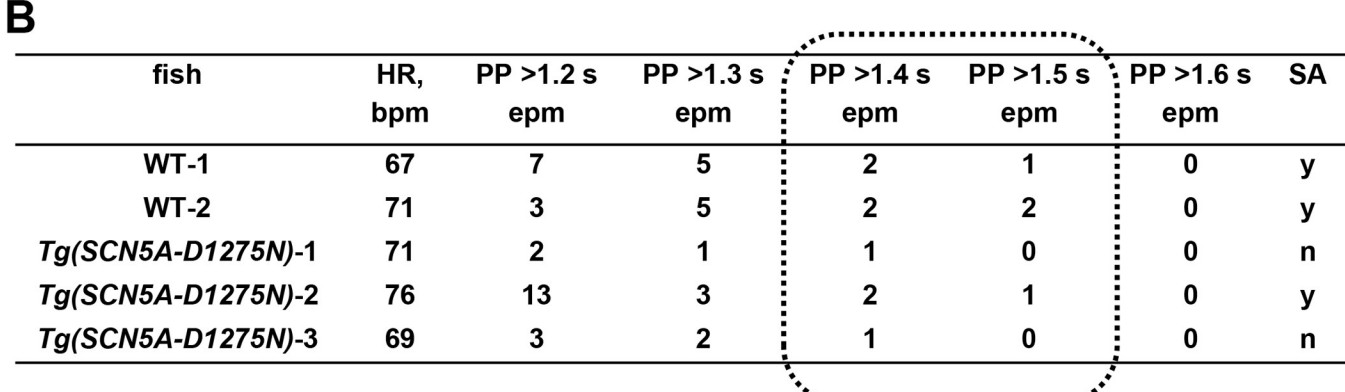
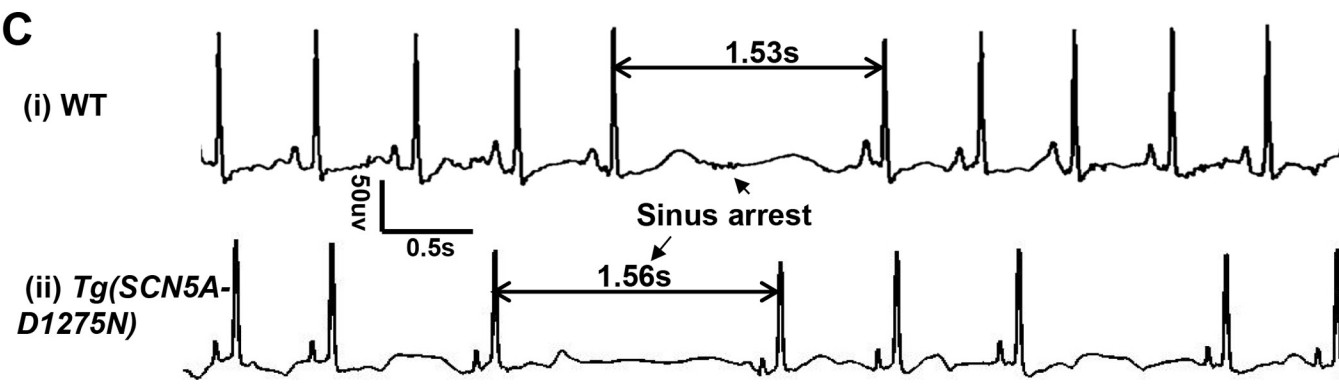

**Fig 4. Evidence supports PP interval more than 1.5 seconds as a criterion to define a Sinus Arrest (SA) episode in adult zebrafish.** A, Summary of electrocardiographic (ECG) data from 59 wild-type (WT) and 16 *Tg(SCN5A-D1275N)* fish at 1 year old. Different cut-offs of PP intervals were used to determine an episode of SA. B, Two wild-type (WT) fish and 3 *Tg(SCN5A-D1275N)* fish that were considered to have SA when 1.2 seconds was used as cut-off, but not if 1.6 seconds was used as cut-off, were further analyzed by reviewing the raw ECG. We considered WT fish 1 and 2 and Tg fish 2 as SA-positive, and others were not because of low frequency of long PP interval. An PP interval more than 1.5 seconds was recommended as a criterion to define SA in zebrafish. C, Representative SA episodes in a WT fish and a *Tg(SCN5A-D1275N)* fish. epm indicates episodes per minute.

adopted more than 1.5 seconds as an arbitrary criterion to define an SA episode (Fig 4B and 4C). On the basis of this criterion, 5.1% of WIK WT fish at 1 year old have SA episodes and 50.0% of *Tg(SCN5A-D1275N)* fish have SA episodes. Consistently, RMSSD, another index that was used to quantify HR variability, was significantly higher in *Tg(SCN5A-D1275N)* fish than WT fish (Fig 5A and Table 1). The average HR is significantly lower, a finding supporting that bradycardia frequently accompanies SA (Fig 4A).

### 3.3 Percentage of WT fish with episodes of SA (WT[SA]) significantly increases during the aging process

Because SSS is an aging-associated disorder in humans, we queried whether the percentage of fish with SA episodes increases in aged fish. We found a remarkable increase of SA in fish at 3 years old compared to 0.5–2 years old (Table 1). Consistently, RMSSD was significantly increased along the aging process (Fig 5A). The average HR decreased gradually (Table 1), as did the P-wave amplitude (Fig 5B). In contrast, QRS complex amplitude showed no obvious difference among the age groups (Fig 5C). As a consequence, the QRS/P ratio of WIK fish at 3 years old was larger than that in younger fish (Fig 5D). Because P-wave amplitude depends on the position of the probes in adult zebrafish (See Fig 1A), we recommend the use of QRS/P ratio as a more reliable index. Interestingly, the frequencies of SA episodes within the subpopulation with SA episodes did not differ significantly in WT fish of different ages or in *Tg(SCN5A-D1275N)* fish (Table 1).

For further investigation, we referred to the subpopulation of all WT fish with SA episodes as WT[SA] and their siblings without SA episodes as WT[Normal]. Similarly, we named the subpopulation of *Tg(SCN5A-D1275N)* fish with SA episodes *Tg(SCN5A-D1275N)*[SA] and their siblings without SA episodes *Tg(SCN5A-D1275N)*[Normal]. In a comparison of WT[SA] and WT[Normal] subgroups, the WT[SA] subgroup had significantly higher RMSSD (Fig 6A), lower HR (Fig 6B), and higher QRS/P ratio (Fig 6C). Similarly, the *Tg(SCN5A-D1275N)*[SA] subgroup had significantly higher RMSSD (Fig 6A) and lower HR (Fig 6B) than the *Tg(SCN5A-D1275N)*[Normal] subgroup, but the QRS/P ratio remained unchanged (Fig 6A).

To exclude the possibility that the increased incidence of SA in the WT[SA] subpopulation during the aging process was specific to the WIK line, we checked TU and NIHGR fish, 2 other WT lines that are maintained in our laboratory. The incidence of SA among different WT fish lines was comparable at the same ages (Table 2). Similarly, the HRs were comparable among different lines at the same ages.

### 3.4 WT[SA] fish manifest defective response to atropine, which is different from that in *Tg(SCN5A-D1275N)*[SA] fish

Given that the increased RMSSD and reduced HR could be a consequence of the presence of SA arrest episodes, additional evidence is needed to prove sinus node dysfunction in WT[SA]. Because chronotropic incompetence is often part of sinus node dysfunction, we assessed whether WT[SA] fish manifested this disability. We tested HR response to atropine, an inhibitor of parasympathetic nerves, by directly applying the solution to the gill. Application of 2 µg/g atropine solution effectively increased HR, and application of 4 µg/g or higher concentration maximized HR (Fig 7A). The maximum HR was reached in 5 minutes and maintained for about 60 minutes (Fig 7B). Therefore, we administered 4 µg/g atropine and measured HR changes at 5 minutes after application. The baseline HR of WT[SA] fish was lower than that in WT[Normal] fish (Fig 7C), and atropine administration did not increase HR in WT[SA] fish to the degree it did in WT[Normal] fish, as indicated by the significant difference in HR between the WT[SA] and WT[Normal] groups (Fig 7C). In contrast, atropine administration increased HR to the maximal level in the *Tg(SCN5A-D1275N)*[SA] fish, although its baseline HR is lower than in *Tg(SCN5A-D1275N)*[Normal] fish (Fig 7D). Together, these results indicate sinus node dysfunction in the WT[SA] fish, supported by the evidence that reduced atropine response occurred in the WT[SA] but not the *Tg(SCN5A-D1275N)*[SA] fish.

### 3.5 WT[SA] fish manifest minor cardiac dysfunction

We determined whether WT[SA] fish hearts manifest any abnormal cardiac structural or functional changes. On high-frequency echocardiography on 9 WT[SA] and 10 WT[Normal] fish, there

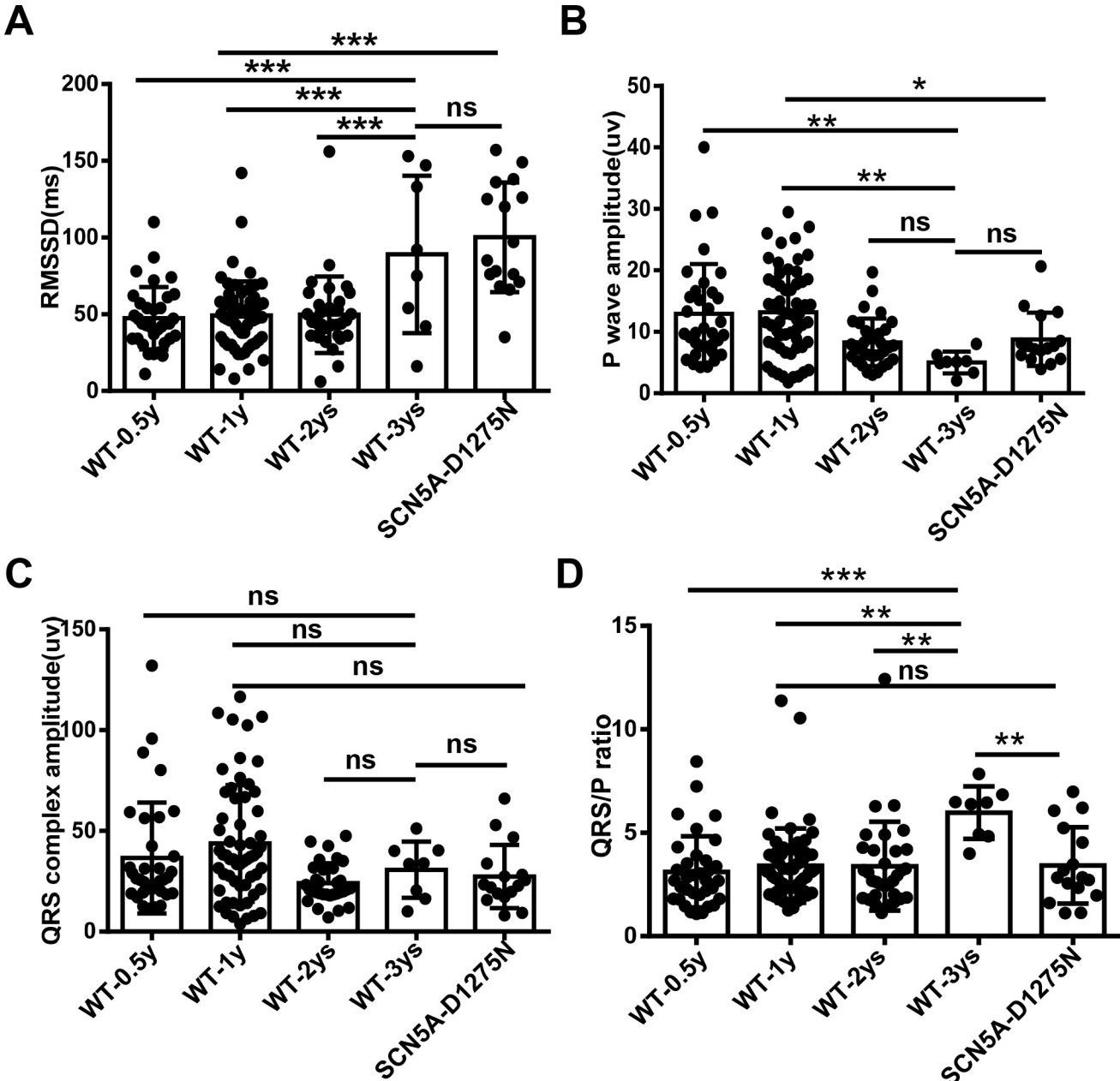

**Fig 5. Root mean square of successive differences (RMSSD) increases, QRS/P ratio increases, and P-wave amplitude reduces during the aging process.** A, RMSSD in wild-type (WT) fish at different ages and in *Tg(SCN5A-D1275N)* fish. B, Quantification of P-wave amplitude in WT fish at different ages and *Tg(SCN5A-D1275N)* fish. C, Quantification of QRS complex amplitude in WT fish at different ages and *Tg(SCN5A-D1275N)* fish. D, Quantification of QRS/P ratio in WT fish at different ages and *Tg(SCN5A-D1275N)* fish. RMSSD and QRS/P ratio are higher, while P wave amplitude is lower, in WT-3 year subpopulation than in other younger siblings. RMSSD, P wave amplitude, QRS/P ratio in the *Tg(SCN5A-D1275N)* fish are higher than those in age-matched WT control. **$P < .01$ and ***$P < .001$ (one-way analysis of variance). ns indicates not significant.

were no significant changes in ejection fraction (EF), end-diastolic volume (EDV)/body weight, end-diastolic volume (EDV)/body weight, ratio between the early (E) ventricular filling velocity and late (A) ventricular filling velocity (E/A ratio), or myocardium performance index (MPI) (Fig 8), but a significantly reduced E wave amplitude and A wave amplitude in WT^SA fish suggested minor cardiac dysfunction (Fig 8F and 8G).

**Table 1. Heart rates and SA incidence comparison in WT and *Tg(SCN5A-D1275N)* transgenic fish.**

| Fish Type | Age | N | Average HR, bpm | Percentage of Fish With SA (No. of Fish) | SA frequency epm |
|---|---|---|---|---|---|
| WT | 0.5y | 34 | 103.5±18.8* | 5.9(2)** | 1.8±0.3 |
| | 1y | 59 | 103.3±15.7* | 5.1(3)** | 2.5±1.2 |
| | 2y | 32 | 103.6±14.2* | 6.3(2)** | 1.5±0.5 |
| | 3y | 8 | 84.6±22.5 | 25.0(2) | 3.0±2.0 |
| Total | | 133 | 101.6±18.4 | 6.8(9) | 2.2±1.3 |
| *Tg(SCN5A-D1275N)* | 1y | 16 | 73.8±14.9### | 50.0(8) ### | 1.9±1.1 |

SA, sinus arrest. epm: episode per minute. HR, heart rate. bpm, beat per minute.

*: p<0.05;

**: p<0.01, Compare with 3 ys WT group;

###: p< 0.001, Compare with 1 y WT group.

## 4. Discussion

### 4.1 Reliable ECG signals can be obtained from adult zebrafish

Compared with existing zebrafish ECG systems that use needle probes invasively inserted into the zebrafish dermis to capture the signal [20, 33, 38, 39], here, we deployed non-invasive electrodes that were applied to the surface of the heart using the commercially available iWORX ECG recording system. We mainly focus on P waves and QRS waves that are sufficient to define SSS. Notably, our methods have not been optimized to reliably monitor T wave. We determined the best position of these probes for generating ECG patterns that resemble those from humans and found that opening the pericardial sac is effective for enhancing the signal and reducing background noise. High-resolution, low-noise single-channel ECG signals can be reliably obtained from more than 90% of fish. A unique application of the ECG recording system we used here does also allow us to document ECG from awake fish, because it is possible to

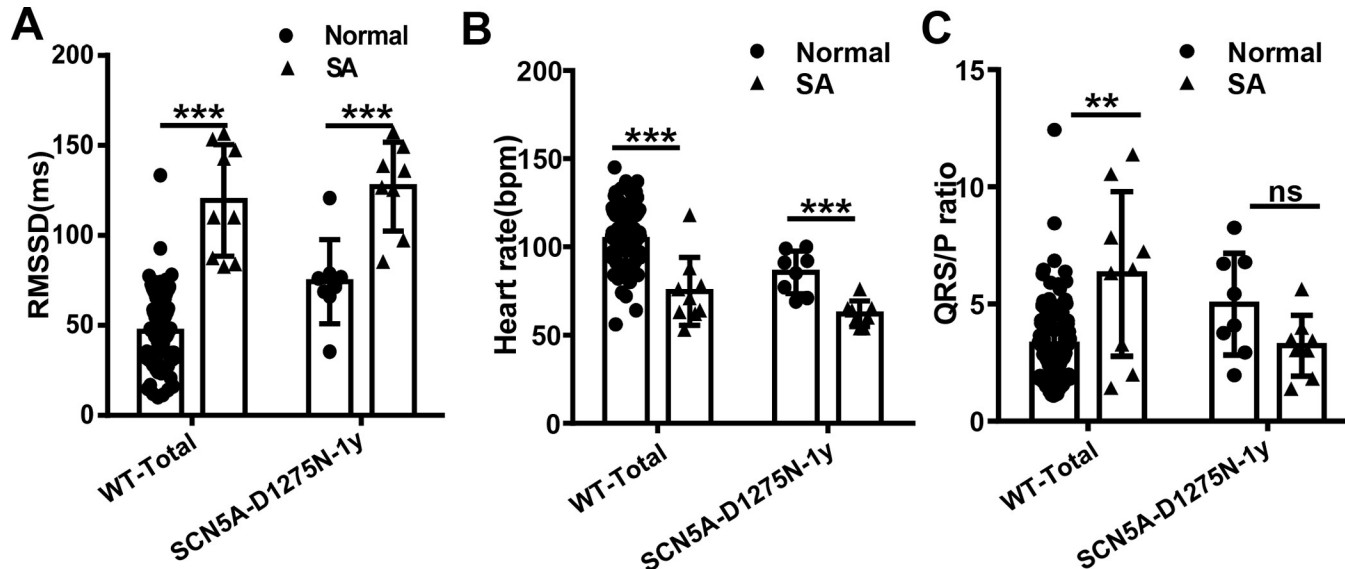

**Fig 6. Obvious features of Sinus Arrest (SA) exist in SA-like subgroup compared with normal fish.** A, Root mean square of successive differences (RMSSD) of wild-type (WT) and *Tg(SCN5A-D1275N)* fish in SA subgroup is higher than in normal fish. B, Heart rate of WT and *Tg(SCN5A-D1275N)* fish in SA subgroup is lower than in normal fish. C, QRS/P ratio of WT and *Tg(SCN5A-D1275N)* fish in SA subgroup is larger than in normal fish. **P < .01 and ***P < .001 (nonparametric tests). bpm indicates beats per minute.

**Table 2. SA incidence in different WT strains.**

| WT strain | Age | N | Average HR, bpm | Percentage of Fish With SA (No. of Fish) | SA frequency epm |
|---|---|---|---|---|---|
| Wik | 1y | 59 | 103.3±15.7 | 5.1(3) | 2.5±1.2 |
| Tu | 1y | 35 | 103.3±12.2 | 0 | 0 |
| Wik | 3y | 8 | 84.6±22.5 | 25.0(2) | 3.0±2.0 |
| NIHGI | 3y | 23 | 81.2±16.7 | 21.7(5) | 2.0±3.0 |

SA, sinus arrest. epm: episode per minute. HR, heart rate. bpm, beat per minute.

physically constrain the fish and prevent body movement for a short time. Nevertheless, we still prefer to anesthetize the zebrafish to obtain more stable and clearer ECG signals.

To reduce experimental variation of the HR, we studied the effect of temperature and noted a nearly linear relationship: an increment in temperature of 1˚C results in an increment of about 10 bpm in zebrafish. As a consequence, HR at 32˚C could be double in the same fish at 22˚C. Therefore, strictly controlled temperature is essential for obtaining reliable ECGs in adult zebrafish [35, 36]. Because a zebrafish facility is typically operated at about 28.5˚C and the room temperature when conducting ECG is usually about 24˚C, we recommend a 1-hour acclimation step before conducting ECG, which could improve data consistency. Of note, because our in-house temperature-controlled chamber might not be ideal, a better temperature-controlled chamber, preferably as a commercially available product, is needed. We did not consider the contribution of evaporation on the body surface to body temperature, which can be considerable when a fish is out of water. An alternative solution is to document ECG signals when a fish is kept within the water, as enabled by ECG jacket technology [40].

Prompted by previous studies that showed significant effects of anesthesia on cardiac function and rhythm [41, 42], we determined the dynamics using the iWorx system. We decided on a 6-minute sedation protocol that ensures deep sleep, as indicated by the effective halt of gills and body movement. Our series of optimization protocols enabled us to obtain reliable ECG signals with consistent results.

### 4.2 Zebrafish can be used to model aging-associated SSS

On the basis of our studies of abnormal heart beating in WT and *Tg(SCN5A-D1275N)* fish, we recommend an PP interval of more than 1.5 seconds as a standard to define an SA episode in adult zebrafish. The use of this arbitrary standard enabled us to quantify SA with software and may make future large-scale analysis of ECG data possible. Given the dramatic effect of body temperature on HR, the use of an absolute PP interval as a criterion must be restricted to a fixed temperature. The criterion needs to be adjusted when ECG is done at temperatures significantly different from 24˚C. Future studies are warranted to determine whether the relative index (ie, the ratio of PP intervals between the 2 neighboring intervals) is a better index than the absolute PP interval to define SA in adult zebrafish. Because the P wave is somewhat variable in a zebrafish ECG, extra cautions need to be taken when an adult fish manifests atrioventricular block that could disengage P waves and R waves.

The establishment of a reliable ECG platform and of a criterion to define an SA episode in an adult zebrafish enabled us to uncover aging-associated SSS in this animal. The percentage of WT fish with SA episodes can be as high as 25% when fish are 3 years old. This subpopulation of WT fish with SA episodes did have other SSS phenotypic traits that are shared with the *Tg(SCN5A-D1275N)* model, a known transgenic fish model for inherited SSS. Bradycardia was found in both models, a finding supporting sinus node dysfunction. However, aging-associated SSS has important differences from the *Tg(SCN5A-D1275N)* model. First, this

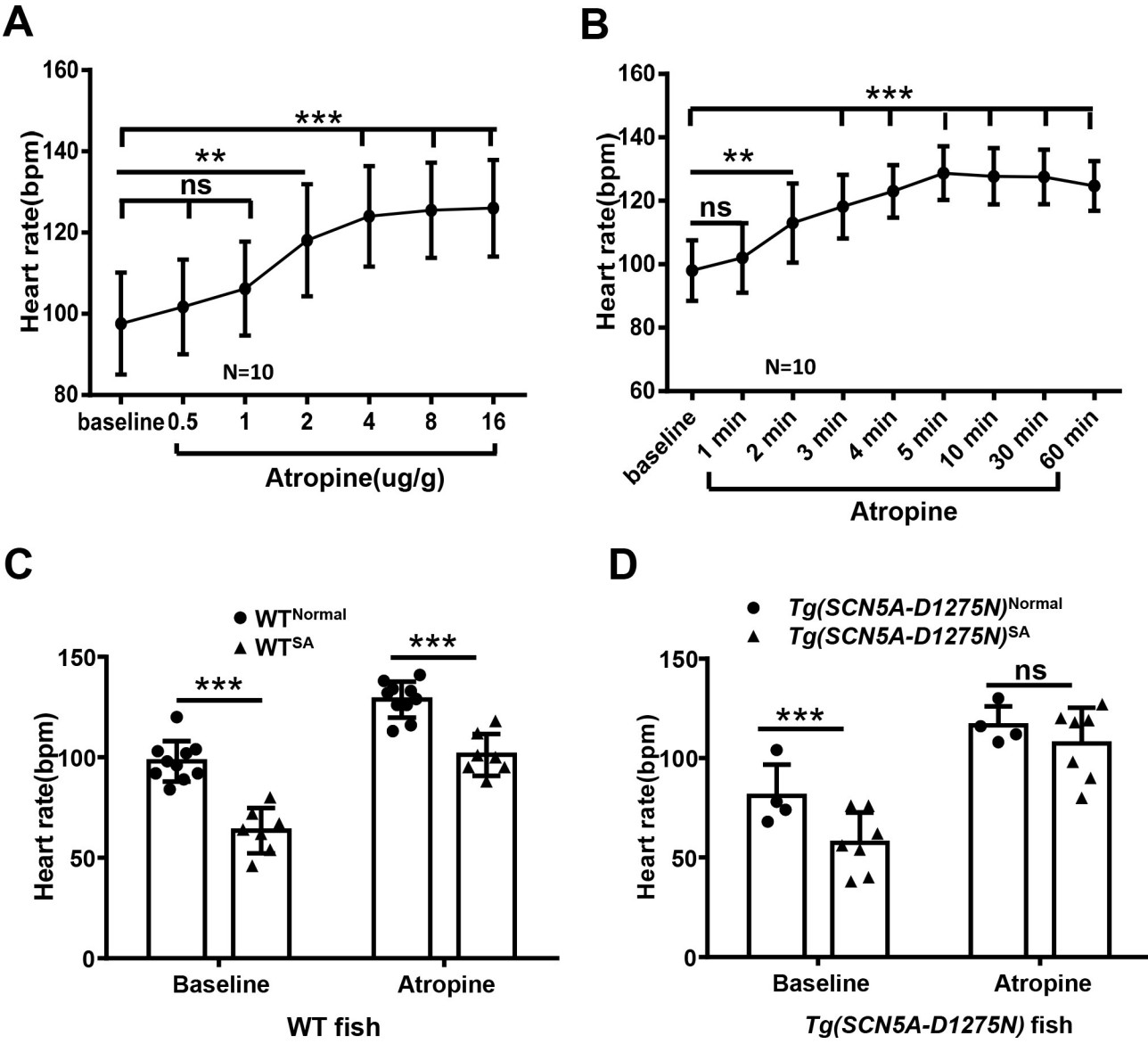

**Fig 7. Chronotropic incompetence in subpopulation of Wild-Type (WT) fish with Sinus Arrest (SA) but not *Tg(SCN5A-D1275N)* fish.** A, Quantification of heart rate in response to different concentrations of atropine (one-way analysis of variance). B, Heart rate response to atropine (4 μg/g) over time (one-way analysis of variance). C, Quantification of heart rate increases in response to atropine (4 μg/g) in WT fish. Heart rate in WT^SA subgroup did not reach maximum, as in WT^Normal siblings. D, Quantification of heart rate increases in response to atropine (4 μg/g) in *Tg (SCN5A-D1275N)* fish. Heart rate in fish with SA episodes reached maximal in their siblings. **P < .01 and ***P < .001 (non-parametric tests). ns indicates not significant.

subpopulation of WT fish has reduced P amplitude and increased QRS/P. Second, this subpopulation of WT fish, but not the *Tg(SCN5A-D1275N)* model, had compromised HR responses to atropine. The data suggested that parasympathetic nerve control of the sinus node is damaged in aging-associated SSS, but not in the transgenic model. These phenotypic differences must reflect different molecular natures of these two SSS models. Similar to aged humans with SSS, fibrosis in the sinus node contributes to the pathogenesis [2, 8], which might explain the increased QRS/P ratio [43]. In contrast, SCN5A is more prominently expressed in the peri-node region, which results in SSS via a different mechanism [44].

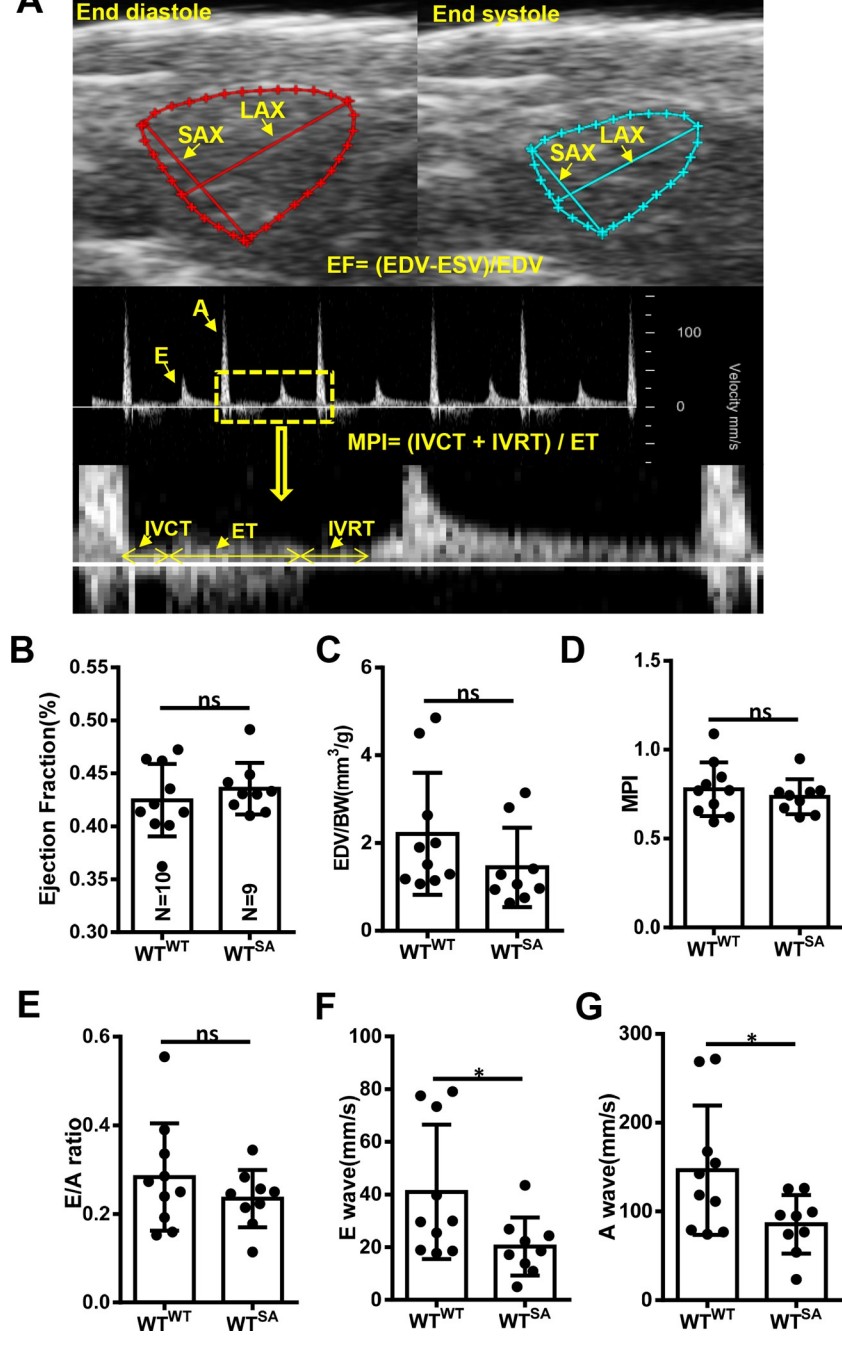

**Fig 8. Cardiac function remains unchanged between wild-type fish with (WT$^{SA}$) and without (WT$^{Normal}$) sinus arrest.** A, Representative B mode echocardiogram of a zebrafish heart at end-diastole (top left) and end-systole (top right) and pulsed-wave Doppler images for quantification of E wave, A wave, IVCT, ET, and IVRT. B-G, Quantification of cardiac function indices between WT$^{Normal}$ and WT$^{SA}$ siblings. $^*P < .05$ (Student $t$ test). A indicates peak velocity of late ventricular filling; BW, body weight; E, peak velocity of early ventricular filling; EDV, end-diastolic volume; EF, ejection fraction; ESV, end-systolic volume; ET, ejection time; IVCT, isovolumic contraction time; IVRT, isovolumic relaxation time; LAX, long axis; MPI, myocardial performance index; ns, not significant; SAX, short axis; WT$^{SA}$, wild-type fish with SA; WT$^{Normal}$, wild-type fish without SA.

### 4.3 A small subpopulation of young WT fish has SA episodes

A notable finding of the study is the identification of a small population of young WT fish that have episodes of SA, when we used an PP interval of more than 1.5 seconds as a standard. Even after we reduce the influence of many confounding factors such as temperature and anesthesia, we still noted a higher variation of HR in adult zebrafish. It is plausible that, compared with mammals, fish lack a certain functional mechanism that ensures stringent control of automacity of heart beating. Whether genetic heterogeneity in the fish WT lines contributes to this phenomenon remains to be clarified. The higher basic incidence in WIK fish than in TU fish tends to support this idea. Possibly, WIK fish have more SSS-related sequence variants than TU fish.

In summary, our data established aging-associated SSS and SA models in adult zebrafish, laying a foundation for discovering underlying genetic factors. A large number of zebrafish mutants have already been generated with either mutagenesis screening or genome editing technology, some of which might affect heart rhythm. The identification of a small subpopulation of WT fish with SA episodes speaks to extra caution when interpreting phenotypes from these mutants. Sufficient numbers of age-matched controls are needed to confidently conclude whether the affected gene is a new genetic factor for SA or SSS.

## Acknowledgments

This work was supported by grants from National Institutes of Health (NIH) R01-HL107304, R01-HL081753, R01-GM63904 and the Mayo Foundation to XX; the Ted and Loretta Rogers Cardiovascular Career Development Award Honoring Hugh C. Smith from Mayo Clinic (FP00093430) to YD, and National Science Foundation (NSF) #1652818 and NIH R41-OD024874 to HC. We thank Beninio Gores and Kashia B. Stragey for managing the zebrafish facility; Dr. Diane Fatkin, Victor Chang Cardiac Research Institute, Australia, for sharing the *Tg(SCN5A-D1275N)* fish.

## Author Contributions

**Conceptualization:** Jianhua Yan, Yonghe Ding, Xiaolei Xu.

**Data curation:** Jianhua Yan, Hongsong Li, Haisong Bu, Kunli Jiao, Alex X. Zhang, Tai Le, Yonghe Ding.

**Formal analysis:** Jianhua Yan, Hongsong Li, Haisong Bu, Kunli Jiao, Tai Le.

**Funding acquisition:** Hung Cao, Xiaolei Xu.

**Investigation:** Jianhua Yan, Yonghe Ding.

**Methodology:** Jianhua Yan, Hongsong Li, Haisong Bu, Kunli Jiao, Alex X. Zhang, Tai Le, Yonghe Ding.

**Project administration:** Xiaolei Xu.

**Supervision:** Yonghe Ding, Xiaolei Xu.

**Validation:** Jianhua Yan.

**Writing – original draft:** Jianhua Yan, Yonghe Ding, Xiaolei Xu.

**Writing – review & editing:** Jianhua Yan, Hung Cao, Yigang Li, Yonghe Ding, Xiaolei Xu.

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
