## [Decision Letter · Decision Letter 0]

25 Feb 2020

PONE-D-20-03005

Aging-Associated Sinus Arrest and Sick Sinus Syndrome in Adult Zebrafish

PLOS ONE

Dear Dr Xu,

Thank you for submitting your manuscript to PLOS ONE. After careful consideration, we feel that it has merit but does not fully meet PLOS ONE’s publication criteria as it currently stands. Therefore, we invite you to submit a revised version of the manuscript that addresses the points raised during the review process.

We would appreciate receiving your revised manuscript by Apr 10 2020 11:59PM. To enhance the reproducibility of your results, we recommend that if applicable you deposit your laboratory protocols in protocols.io, where a protocol can be assigned its own identifier (DOI) such that it can be cited independently in the future. For instructions see: http://journals.plos.org/plosone/s/submission-guidelines#loc-laboratory-protocols

We look forward to receiving your revised manuscript.

Kind regards,

Andrea Barbuti, PhD

Academic Editor

PLOS ONE

Journal Requirements:

1. Please include your tables as part of your main manuscript and remove the individual files. Please note that supplementary tables (should remain/ be uploaded) as separate "supporting information" files

Reviewers' comments:

Reviewer's Responses to Questions

**Comments to the Author**

1. Is the manuscript technically sound, and do the data support the conclusions?

Reviewer #1: Yes

Reviewer #2: Yes

2. Has the statistical analysis been performed appropriately and rigorously? 

Reviewer #1: Yes

Reviewer #2: Yes

3. Have the authors made all data underlying the findings in their manuscript fully available?

Reviewer #1: Yes

Reviewer #2: Yes

4. Is the manuscript presented in an intelligible fashion and written in standard English?

Reviewer #1: Yes

Reviewer #2: Yes

5. Review Comments to the Author

Reviewer #1: Aging-Associated Sinus Arrest and Sick Sinus Syndrome in Adult Zebrafish

In this manuscript the authors aim to define Zebrafish as a new model for aging-associated Sick Sinus Syndrome. They first describe an optimization procedure for measuring ECG signals in anesthetized Zebrafish; they define a time threshold of PP interval for the determination of Sinus arrest episodes in Zebrafish; they evaluate the quantity of SA episodes during the aging process and they show a higher incidence in old fishes; they finally show a difference in the response to Atropine in young and old fishes.

Here are some points to take in consideration:

Major points:

INTRODUCTION

Lines 43-48: please give more details about the sinoatrial node (SAN), intrinsic heart rate and the autonomic nervous system.

Before introducing the index for diagnosis of SSS, the possible causes of SSS should be mentioned , like fibrosis, remodeling of the SAN and ion channels dysfunction.

Lines 84-85: the authors didn’t mention these SSS phenotypic traits before, but they just mentioned PP interval. It’s not clear if they are just present in the Zebrafish model or how they relate to humans. Please explain these indexes and add citations.

METHODS

Line 99: at which pH?

Line 106: is this the time for induction of anesthesia? How long does this anesthesia last?

RESULTS

Lines 172-178: the authors state they optimized the recordings by finding the best position for the ECG electrodes. This investigation was already done by other authors (Crowcombe 2016; Zhao 2019). In this previous work the T wave was positive, while in this work it’s not clear. It would be nice if the authors could identify more efficiently the T wave and explain in which way their findings represent an improvement.

Line 197: Why do you have an increase in HR after injection of anesthesia? Is it just due to stress induced by the injection? Please comment.

Line 203: in the methods they say they leave the fish for 3 minutes in the E3 water with Tricaine; here it seems they leave them for 6 minutes. It’s confusing. It would be better if in the methods they could explain better the protocol they use and how Tricaine works.

Line 204: the recordings are performed 7 minutes after induction of anesthesia. Is this sufficient to reach the steady state after the stress associated with the injection procedure? From the figure it seems that after 6 minutes HR is still decreasing. And, again, for how long does the anesthesia last?

Lines 212-213: are they comparing HR in normal conditions or under anesthesia? From the manuscripts it’s not clear.

Line 244: is it the P-wave amplitude a safe parameters to consider? As the authors mentioned previously, a movement in the positioning of the electrodes could change the amplitude of the waves; this could influence the results.

Lines 250-251: it’s not clear if they now call WTSA the 3 year old-fishes of or all the WT fishes with SA.

Lines 254-255: The higher RMSSD and the lower HR could just be a consequence of the presence of SA arrest episodes that are included in the calculation of the means of these parameters.

How do the authors consider regulation by the autonomic system?

Lines 265-266: Please calculate intrinsic heart rates of 1 year-old and 3 year-old fishes. And then it would be possible to use this parameter as a starting point to check responses to atropine. Without this we don’t know the reference value for the comparison.

Lines 277-278: they state that these data indicate a sinus node dysfunction in WTSA fishes supported by the fact that response to Atropine is different from the one found in the model for SSS. This is controversial.

Line 497: this legend title is confusing. It gives a negative connotation of the use of anesthesia but throughout the work anesthesia is used because it gives better ECGs.

Figure 1 B: it is not clear which wave is the T wave. And in any case the 2 ECGs look quite different from the one they define as the correct one in Figure 1A.

Figure 2 A: between 22°C and 26°C there seems to be a higher beat to beat variability. Why? The authors don’t mention anything about it.

Figure 2 B: please comment the high difference between heart rate in presence or absence of Tricaine.

Figure 2C: in the methods the authors say that they anesthetize the mice for 3 minutes and then move to the chamber for recordings. How did they do the recordings after 1.5 minutes? Is the induction of anesthesia enough? Or probably they did the measurements 1.5 minutes after the 3 minutes of incubation with Tricaine. It’s not completely clear.

Figure 3: it is not clear whether they are comparing Zebrafish under anesthesia to human without. In this case it would make no sense. If it isn’t like this, please specify it.

I would also prefer to see a comparison of ECGs with some previous published recordings in Zebrafish.

Figure 5A: the variability among 3-year old fishes it’s really high. Increasing of the number of experiments would help.

DISCUSSION

Lines 295-297: the authors consider the recordings from awake fishes. They should explain why at the end they chose to record under anesthesia. From the results it seems that it is because of lower noise. But on the other hand a measure without anesthesia could have some advantages. Please discuss .

Line 334: It’s not possible to conclude from their results if it is a bradycardia due to intrinsic problems or to autonomic imbalance.

Line 349: the high HR variation could just be due to body temperature decrease during experiments.

Minor points:

Line 60: “the most prevalent genes involved in congenital SSS”.

Line 88: The authors write “Methods and Methods”. They probably mean “Materials and Methods”.

Line 167: mean ± SD.

Line 214: Figure 3B-C.

Reviewer #2: In their manuscript, Yen et al. present a thorough and carefully carried out analysis on the heart rhythm of adult zebrafish. The authors first optimized the protocol for the measurement of electrocardiogram using the commercially available iWork system. They obtained stronger and more consistent ECG signals by opening the pericardial sac and controlling temperature of the local environment. Using this improved protocol, the authors examined the heart rate of wild type fish and Tg(SCN5A-D1275N), an established zebrafish model for inherited sick sinus syndrome model. Using a criterion of PP interval > 1.5 seconds, approximately half of the Tg(SCN5A-D1275N) fish are defined as having SA episodes. More interestingly, the authors noted an increased percentage of wild type fish with SA episodes as they age. Furthermore, using the anticholinergic drug atropine, the authors demonstrate different sick sinus symptom-related traits between Tg(SCN5A-D1275N) and wild type aged fish. The authors conclude that zebrafish might serve as an alternative model for the study of aging-associated sick sinus syndrome. Overall, the manuscript is clearly written. The data are of high-quality and the improved protocol for ECG recording will help establish zebrafish as an animal model for the study of sinus dysfuntion.

Specific Comments:

1. It is surprising to note that the heart rate of adult zebrafish is quite variable based on the authors’ measurement. Could the authors comment on how their data compared to previous reports? It would also be very helpful if the authors could comment on what might contribute to the irregular heartbeat detected in their recording. The traces shown in Fig.2A illustrate that the heart rate is more irregular when recorded at lower temperatures (22 -26oC v.s. 30-32oC). Would the temperature or the surgery contribute to the high variability in heart rate?

2. The ECG analysis identified a subpopulation of wild type and Tg(SCN5A-D1275N) fish with SA episodes. Given the irregular heartbeat observed in wild type and Tg(SCN5A-D1275N) fish, it would be helpful for the authors to clarify whether WTSA and Tg(SCN3A-D1275N)SA fish consistently expressing SA episodes. Also, have the authors re-examined WTnormal and Tg(SCN3A-D1275N)normal fish? Do some of them exhibit SA when re-examined?

3. The authors showed that atropine treatment increases the heart rate of Tg(SCN3A-D1275N)SA to the level of Tg(SCN3A-D1275N)normal fish, but the heart rate of atropine-treated WTSA fish is still slower than atropine-treated WTnormal. The authors conclude that there is a reduced atropine response occurred in the WTSA but not the Tg(SCN3A-D1275N)SA fish. However, atropine treatment appears to increase the heart rate in both WTSA and WTnormal fish (Fig.7C). Please clarify.

Other comments:

1. Figure 4A, percentage of 3yr-old wild type fish with long PP interval should be 62.5% for PP>1.2 s, 25% for PP> 1.5 S and 25% for PP>1.6 s.

2. Page 15, line 286, it should be Figure 8 F and G.

3. Table 2 is missing.

6. PLOS authors have the option to publish the peer review history of their article (what does this mean?). If published, this will include your full peer review and any attached files.

Reviewer #1: No

Reviewer #2: No

---

## [Author Response · Author response to Decision Letter 0]

31 Mar 2020

Below please find our point-to-point answers. 

Reviewer #1: Aging-Associated Sinus Arrest and Sick Sinus Syndrome in Adult Zebrafish

In this manuscript the authors aim to define Zebrafish as a new model for aging-associated Sick Sinus Syndrome. They first describe an optimization procedure for measuring ECG signals in anesthetized Zebrafish; they define a time threshold of PP interval for the determination of Sinus arrest episodes in Zebrafish; they evaluate the quantity of SA episodes during the aging process and they show a higher incidence in old fishes; they finally show a difference in the response to Atropine in young and old fishes.

Here are some points to take in consideration:

Major points:

INTRODUCTION

Lines 43-48: please give more details about the sinoatrial node (SAN), intrinsic heart rate and the autonomic nervous system.

Thanks to your suggestion, we made the following changes (new lines 44-49): “Heart rhythm is primarily initiated by the automatic beating of pacemaker cells located in the sinoatrial node (SAN), a specialized area in the upper right chamber of a mammalian heart. The initial innate electrical potential transmits from the SAN to the atrioventricular node (AVN) and finally passes to the His-Purkinje system. This well-controlled rhythmic contraction is modulated both positively by sympathetic nerves and negatively by parasympathetic nerves [1]”.

Before introducing the index for diagnosis of SSS, the possible causes of SSS should be mentioned, like fibrosis, remodeling of the SAN and ion channels dysfunction.

Thanks to your suggestion, we added the following sentences to mention the possible causes of SSS (new lines 51-55): “The causes of SSS can be divided into both intrinsic and extrinsic factors that disrupt the SAN function. Intrinsic causes include age-related degenerative fibrosis, ion channel dysfunction, and remodeling of the SAN. Extrinsic factors include the use of certain pharmacologic agents, metabolic disturbances, and autonomic dysfunctions that exacerbate SSS”.

Lines 84-85: the authors didn’t mention these SSS phenotypic traits before, but they just mentioned PP interval. It’s not clear if they are just present in the Zebrafish model or how they relate to humans. Please explain these indexes and add citations.

We now add sentences to mention these phenotypic traits. We also include related citations to show these indices are presented in both Zebrafish and humans (new lines: 59-61, and 91-92)

METHODS

Line 99: at which pH?

pH 7.0. It is now added to the text (new line: 107).

Line 106: is this the time for induction of anesthesia? How long does this anesthesia last?

Yes, this is the time (3 minutes) we used to induce anesthesia before we start to operate microsurgery for opening the pericardium sac. The anesthesia usually lasted for additional 1-2 minutes, which is sufficient for us to complete the surgery. After the surgery, fish were returned to tank water for recovery. We now clarified this protocol in the text (new lines 114-115, 118-119)

RESULTS

Lines 172-178: the authors state they optimized the recordings by finding the best position for the ECG electrodes. This investigation was already done by other authors (Crowcombe 2016; Zhao 2019). In this previous work the T wave was positive, while in this work it’s not clear. It would be nice if the authors could identify more efficiently the T wave and explain in which way their findings represent an improvement.

Other published work mostly used probes that were invasively inserted into the zebrafish dermis to capture the signal. In contrast, we deployed non-invasive electrodes that were applied to the surface of the heart using the iWORX ECG recording system. In this manuscript, we mainly focus on P waves and QRS waves that are sufficient to define SSS. Our methods have not been optimized to reliably monitor T wave. Thanks to your suggestion, we added these two references, and described the limitation of our methods in the discussion section (new lines 307-312). 

Line 197: Why do you have an increase in HR after injection of anesthesia? Is it just due to stress induced by the injection? Please comment.

This is an interesting discovery that we do not have full explanation. Our speculation is that the increased HR by tricaine is a consequence of either stress response or excessive excitement of sympathetic nerves. Notably, we didn’t inject tricaine to induce anesthesia. We meant “incubation” by “administration”. To avoid confusion, we now use the word “incubation”. We added our speculation to the text (new lines 207).

Line 203: in the methods they say they leave the fish for 3 minutes in the E3 water with Tricaine; here it seems they leave them for 6 minutes. It’s confusing. It would be better if in the methods they could explain better the protocol they use and how Tricaine works.

We used different anesthesia times of tricaine in different experiments. For microsurgery, we used 3 minutes of tricaine incubation. For ECG and Echo, fish need to be more deeply anesthetized; thus, we used a longer anesthesia time (6 minutes). We clarified these points in the method section (lines 114-115, 133). 

Line 204: the recordings are performed 7 minutes after induction of anesthesia. Is this sufficient to reach the steady state after the stress associated with the injection procedure? From the figure it seems that after 6 minutes HR is still decreasing. And, again, for how long does the anesthesia last?

Indeed, the heart rate continued to reduce after 6 minutes of anesthesia, but not significantly. Comparing to earlier time points, we considered that fish reach relatively steady state after 6 minutes of anesthesia. Indeed, more reliable ECG signals can be recorded. The anesthesia usually lasts additional 2 minutes that enables us to obtain reliable ECG recording (see Figure 2E). We added these details to the text (new lines 216-217).

Lines 212-213: are they comparing HR in normal conditions or under anesthesia? From the manuscripts it’s not clear.

Sorry for the confusion. We meant under anesthesia. We added this premise accordingly (new line 226).

Line 244: is it the P-wave amplitude a safe parameters to consider? As the authors mentioned previously, a movement in the positioning of the electrodes could change the amplitude of the waves; this could influence the results.

We agree. We deleted statement that the P-wave amplitude alone can serve as “an index indicating a decline in sinus node function” and used only the QRS/P ratio instead as a more reliable index (new lines 257-261). 

Lines 250-251: it’s not clear if they now call WTSA the 3 year old-fishes of or all the WT fishes with SA.

Sorry for the confusion. WTSA refers to all the wild-type fish with SA (new lines 264-265).

Lines 254-255: The higher RMSSD and the lower HR could just be a consequence of the presence of SA arrest episodes that are included in the calculation of the means of these parameters.

How do the authors consider regulation by the autonomic system?

We agree that there is a possibility that these two indices could be simply a consequence of the presence of SA arrest episodes. Indeed, this was the reason that prompted us to seek additional evidences by checking the autonomic system in section 3.4., which lead us to identify reduced response to atropine. We added this rationale to the text (new line 2791-281).

Lines 265-266: Please calculate intrinsic heart rates of 1 year-old and 3 year-old fishes. And then it would be possible to use this parameter as a starting point to check responses to atropine. Without this we don’t know the reference value for the comparison.

As a pilot study, we have tested different doses of atropine to identify the maximal dose that completely remove the influence of parasympathetic nerve system. We then detected uniquely reduced response in WTSA that is different from WTnormal and Tg(SCN5)SA, which strongly supports a defective SAN in WTSA. In the literature, it appears acceptable to define autonomic dysfunction in SSS by only manipulating parasympathetic nerves system (Vavetsi S et al. Europace. 2008). We agree that IHR is an interesting index to define in adult zebrafish. However, we plan to measure it in our future studies. 

Vavetsi S, Nikolaou N, Tsarouhas K, Lymperopoulos G, Kouzanidis I, Kafantaris I, Antonakopoulos A, Tsitsimpikou C, Kandylas J.Consecutive administration of atropine and isoproterenol for the evaluation of asymptomatic sinus bradycardia. Europace. 2008;10(10):1176-81. doi: 10.1093/europace/eun211. Epub 2008/08/13. PubMed PMID: 18701603.

Lines 277-278: they state that these data indicate a sinus node dysfunction in WTSA fishes supported by the fact that response to Atropine is different from the one found in the model for SSS. This is controversial.

Because reduced atropine response only occurs in WTSA fish, we reasoned that sinus node dysfunction in these fish is likely characterized with aberrant parasympathetic control. By contrast, Tg(SCN5)SA fish does not manifest this phenotypic trait, suggesting a different SSS mechanism. We clarified this statement further in our discussion section (new lines 359-363). 

Line 497: this legend title is confusing. It gives a negative connotation of the use of anesthesia but throughout the work anesthesia is used because it gives better ECGs.

Thanks for your suggestion. We agree and now change the legend title to “Optimization of temperature and anesthesia for reliable ECG recording in adult zebrafish” (new lines 551-552).

Figure 1 B: it is not clear which wave is the T wave. And in any case the 2 ECGs look quite different from the one they define as the correct one in Figure 1A.

We acknowledge that our current protocol is mainly opted to study P and QRS waves, but have not been optimized to reliably define the T-wave. We added this point to the text (new line 311-312).

Figure 2 A: between 22°C and 26°C there seems to be a higher beat to beat variability. Why? The authors don’t mention anything about it.

Figure 2 B: please comment the high difference between heart rate in presence or absence of Tricaine.

Thanks for pointing out this potentially highly interesting phenomenon. We now described this observation in the text and indicated that it is warranted for further quantification and further study in the future (new lines 202-204).

We now comment the high difference between heart rate in presence or absence of tricaine (new lines 209-211). 

Figure 2C: in the methods the authors say that they anesthetize the mice for 3 minutes and then move to the chamber for recordings. How did they do the recordings after 1.5 minutes? Is the induction of anesthesia enough? Or probably they did the measurements 1.5 minutes after the 3 minutes of incubation with Tricaine. It’s not completely clear.

Sorry for the confusion. The times indicated in this Figure are anesthesia times that we used before we move the fish to the chamber for ECG documentation. 

Indeed, 1.5 minutes seems not sufficient, as indicated in Figure 2C, and we recommend 6 minutes of anesthesia, as shown in Figure 2E.

Figure 3: it is not clear whether they are comparing Zebrafish under anesthesia to human without. In this case it would make no sense. If it isn’t like this, please specify it.

In this case, we intended compare ECG from fish, mouse and human under anesthesia. We replaced the previous ECG image of an awaken patient with a new ECG image of a patient under anesthesia. We now specify this fact in the text (new lines 560-561). 

I would also prefer to see a comparison of ECGs with some previous published recordings in Zebrafish.

We added references for several previous published recordings, which stated similar HR (new lines 221-222). 

Figure 5A: the variability among 3-year old fishes it’s really high. Increasing of the number of experiments would help.

We agree that the RMSSD index in 3-year old WT-wik fish is high. Unfortunately, we don’t have more WT-wik fish for this experiment. Nevertheless, the statistic power is sufficient for us to reach our conclusion. 

DISCUSSION

Lines 295-297: the authors consider the recordings from awake fishes. They should explain why at the end they chose to record under anesthesia. From the results it seems that it is because of lower noise. But on the other hand a measure without anesthesia could have some advantages. Please discuss.

Thanks for this great suggestion. We now discuss the underlying rational in more detail (new lines 318-319): anesthesia affects the heart rate, but to obtain stable and clear ECG signals, zebrafish need to be fully anesthetized.

Line 334: It’s not possible to conclude from their results if it is a bradycardia due to intrinsic problems or to autonomic imbalance.

Agree. Our current data cannot completely discern these two factors. However, our atropine experiment did unveil a contribution from autonomic imbalance. 

Line 349: the high HR variation could just be due to body temperature decrease during experiments.

Indeed, our manuscript aims to alert the readers on the huge impacts of temperature on ECG in adult zebrafish. We agree that a better temperature-controlled chamber is needed to further reduce HR variation. We added this point to the text and changed “eliminate” to “reduce the influence of“(new lines 371-372).

Minor points:

Line 60: “the most prevalent genes involved in congenital SSS”.

We changed to “two well-established causative genes for SSS” (new lines 67-68).

Line 88: The authors write “Methods and Methods”. They probably mean “Materials and Methods”.

Sorry for the typo. We corrected it to Materials and Methods (new line 96).

Line 167: mean ± SD.

Thank you for the correction. We edited it accordingly (new line 178).

Line 214: Figure 3B-C.

Thank you for the correction. We edited it accordingly (new lines 227).

Reviewer #2: In their manuscript, Yen et al. present a thorough and carefully carried out analysis on the heart rhythm of adult zebrafish. The authors first optimized the protocol for the measurement of electrocardiogram using the commercially available iWork system. They obtained stronger and more consistent ECG signals by opening the pericardial sac and controlling temperature of the local environment. Using this improved protocol, the authors examined the heart rate of wild type fish and Tg(SCN5A-D1275N), an established zebrafish model for inherited sick sinus syndrome model. Using a criterion of PP interval > 1.5 seconds, approximately half of the Tg(SCN5A-D1275N) fish are defined as having SA episodes. More interestingly, the authors noted an increased percentage of wild type fish with SA episodes as they age. Furthermore, using the anticholinergic drug atropine, the authors demonstrate different sick sinus symptom-related traits between Tg(SCN5A-D1275N) and wild type aged fish. The authors conclude that zebrafish might serve as an alternative model for the study of aging-associated sick sinus syndrome. Overall, the manuscript is clearly written. The data are of high-quality and the improved protocol for ECG recording will help establish zebrafish as an animal model for the study of sinus dysfuntion.

Specific Comments:

1. It is surprising to note that the heart rate of adult zebrafish is quite variable based on the authors’ measurement. Could the authors comment on how their data compared to previous reports? It would also be very helpful if the authors could comment on what might contribute to the irregular heartbeat detected in their recording. The traces shown in Fig.2A illustrate that the heart rate is more irregular when recorded at lower temperatures (22 -26oC v.s. 30-32oC). Would the temperature or the surgery contribute to the high variability in heart rate?

Indeed, a major discovery of this manuscript is to report high variation of HR in an adult zebrafish. The phenomenon was not emphasized in previously published papers, many of which might deem the phenomenon as a confounding factor. Here, we provided strong data to demonstrate that this variation is not random, needs to be considered, and is associated with aging. Our data prompts future studies to uncover underlying mechanisms (new lines 381-384).

Thanks for pointing out that the heart rate is more irregular when recorded at lower temperatures. We now described this observation in the text (lines 202-204), and indicated that this phenomenon shall be quantified and further studied as a future direction.

2. The ECG analysis identified a subpopulation of wild type and Tg(SCN5A-D1275N) fish with SA episodes. Given the irregular heartbeat observed in wild type and Tg(SCN5A-D1275N) fish, it would be helpful for the authors to clarify whether WTSA and Tg(SCN3A-D1275N)SA fish consistently expressing SA episodes. Also, have the authors re-examined WTnormal and Tg(SCN3A-D1275N)normal fish? Do some of them exhibit SA when re-examined?

Yes, we have re-examined WTSA and Tg(SCN3A-D1275N)SA fish at different times, and we did find that they manifest SA episodes consistently. It is highly possible that some WTnormal might have SA episodes, if the sensitivity is increased by monitoring ECG with a longer time frame. 

3. The authors showed that atropine treatment increases the heart rate of Tg(SCN3A-D1275N)SA to the level of Tg(SCN3A-D1275N)normal fish, but the heart rate of atropine-treated WTSA fish is still slower than atropine-treated WTnormal. The authors conclude that there is a reduced atropine response occurred in the WTSA but not the Tg(SCN3A-D1275N)SA fish. However, atropine treatment appears to increase the heart rate in both WTSA and WTnormal fish (Fig.7C). Please clarify.

Yes, atropine treatment increased heart rates in both WTSA and WTnormal fish, as well as Tg(SCN3A-D1275N)SA and Tg(SCN3A-D1275N)normal fish. However, the rate of their increase is significantly different, suggesting defective autonomic response only occurs in WTSA fish. 

Other comments:

1. Figure 4A, percentage of 3yr-old wild type fish with long PP interval should be 62.5% for PP>1.2 s, 25% for PP> 1.5 S and 25% for PP>1.6 s.

Thank you very much for pointing out these errors. We now correct them accordingly.

2. Page 15, line 286, it should be Figure 8 F and G.

Thank you very much. We now correct the typo (new line 303).

3. Table 2 is missing.

We now add back Table 2.

---

## [Decision Letter · Decision Letter 1]

16 Apr 2020

Aging-Associated Sinus Arrest and Sick Sinus Syndrome in Adult Zebrafish

PONE-D-20-03005R1

Dear Dr. Xu,

We are pleased to inform you that your manuscript has been judged scientifically suitable for publication and will be formally accepted for publication once it complies with all outstanding technical requirements.

With kind regards,

Andrea Barbuti, PhD

Academic Editor

PLOS ONE

Additional Editor Comments (optional):

Reviewers' comments:

Reviewer's Responses to Questions

**Comments to the Author**

1. If the authors have adequately addressed your comments raised in a previous round of review and you feel that this manuscript is now acceptable for publication, you may indicate that here to bypass the “Comments to the Author” section, enter your conflict of interest statement in the “Confidential to Editor” section, and submit your "Accept" recommendation.

Reviewer #1: All comments have been addressed

Reviewer #2: All comments have been addressed

2. Is the manuscript technically sound, and do the data support the conclusions?

Reviewer #1: Yes

Reviewer #2: Yes

3. Has the statistical analysis been performed appropriately and rigorously? 

Reviewer #1: Yes

Reviewer #2: Yes

4. Have the authors made all data underlying the findings in their manuscript fully available?

Reviewer #1: Yes

Reviewer #2: (No Response)

5. Is the manuscript presented in an intelligible fashion and written in standard English?

Reviewer #1: Yes

Reviewer #2: (No Response)

6. Review Comments to the Author

Reviewer #1: The authors sufficiently addressed al my points. From my side the study is sound now. I recommend to accept the paper now.

Reviewer #2: (No Response)

7. PLOS authors have the option to publish the peer review history of their article (what does this mean?). If published, this will include your full peer review and any attached files.

Reviewer #1: No

Reviewer #2: No

---

## [Editor Report · Acceptance letter]

17 Apr 2020

PONE-D-20-03005R1 

Aging-Associated Sinus Arrest and Sick Sinus Syndrome in Adult Zebrafish 

Dear Dr. Xu:

I am pleased to inform you that your manuscript has been deemed suitable for publication in PLOS ONE. Congratulations! Your manuscript is now with our production department. 

With kind regards,

on behalf of

Dr. Andrea Barbuti 

Academic Editor

PLOS ONE